# Biological Treatments for Pediatric Psoriasis: State of the Art and Future Perspectives

**DOI:** 10.3390/ijms231911128

**Published:** 2022-09-22

**Authors:** Federico Diotallevi, Oriana Simonetti, Giulio Rizzetto, Elisa Molinelli, Giulia Radi, Annamaria Offidani

**Affiliations:** Dermatological Clinic, Department of Clinical and Molecular Sciences, Polytechnic Marche University, 60126 Ancona, Italy

**Keywords:** psoriasis, pediatric psoriasis, biological treatments, biologics

## Abstract

Psoriasis is a chronic systemic inflammatory disease that primarily affects the skin and is associated with multiple comorbidities with a considerable reduction in quality of life of affected patients. One-third of psoriasis cases begin in childhood and are associated with significant medical comorbidities such as obesity, metabolic syndrome, arthritis, and psychiatric disorders. In addition, because of its chronic nature and frequent relapses, psoriasis tends to require long-term treatment. Treatment of pediatric psoriasis usually involves the same methods used for adults. However, most treatments for pediatric psoriasis are used off-label, and research in this regard is still lacking. Targeted therapies involving the use of newly developed biologic drugs are also increasingly being applied to childhood psoriasis. This review summarizes the clinical features of pediatric psoriasis and focuses mainly on the updated concepts of pathogenesis and biological treatments of pediatric psoriasis.

## 1. Introduction

Psoriasis (PsO) is a multifactorial immune-mediated inflammatory disease, with a chronic relapsing remitting course, which affects 2–3% of the worldwide population. It is characterized by aberrant T cell activity causing hyperproliferation of epidermal keratinocytes, with development of erythematous skin plaques covered by a silvery or micaceous scale [1,2]. Plaque psoriasis, also known as “psoriasis vulgaris”, is the most common variant of psoriasis and is characterized by plaques mainly localized in the elbows, knees, scalp and trunk [3]. One-third of psoriasis cases start during childhood [4]. Pediatric psoriasis differs from adult-onset psoriasis in types of environmental triggers, with trauma, stress, and bacterial infection being the most common pediatric disease triggers, while drug reactions, smoking, alcohol use, and underlying HIV infection are triggers more common in adulthood [5]. Recently, the recognition of psoriasis as a systemic inflammatory disease in which cutaneous involvement represents the most clinically evident part of a complex underlying pathogenic mechanism has been linked to both pediatric and adult psoriasis [6,7]. In fact, more in children than in adults, psoriasis appears to be associated with several comorbidities, such as obesity and other cardiovascular risk factors, arthritis, and psychiatric disorders [8].

Treatment options available today are the result of years of research into the pathogenesis of the disease which led to the definition of treatments with increasingly specific molecular targets resulting in high efficacy and a reduced number of side effects [9,10,11].

The aim of this paper is, through a narrative review of the literature, to analyze the pathogenetic features of pediatric psoriasis and the latest biologic treatments available for the management of this condition.

## 2. Materials and Methods

This narrative review was based on the general approach of the biomedical narrative review construction that involves four key steps: 1—identify keywords; 2—conduct research; 3—review abstract and articles; 4—document results [12].

### 2.1. Identify Keywords

To identify keywords, a brainstorming approach, involving the entire research group, was used. The research team consisted of two dermatologists with expertise in psoriasis pathophysiology, two dermatologists with expertise in psoriasis therapy, and two dermatologists with expertise in pediatric psoriasis. Two among them also had specific expertise in literature review methodology. During the first meeting, the research team selected the topic, identified the scope, constructed the title, and selected the keywords as follows: “pediatric psoriasis”, “pediatric psoriasis AND pathophysiology”, “pediatric psoriasis AND therapy”, pediatric psoriasis AND treatment”, “pediatric psoriasis AND biologic treatment”.

### 2.2. Conduct Research

An extensive search for eligible articles was conducted on the following databases: National Library of Medicine PubMed, and Scopus. The references list of selected studies was scanned to find additional records. Inclusion criteria: studies reporting on pediatric psoriasis, published in the English language, published between 1990 and 2020, abstract available. No restriction on the design of the study was considered, and randomized controlled trials, case-control studies, cross-sectional studies, case reports and series, and review articles were included.

### 2.3. Review Abstract and Article

The selection of the relevant data published in the literature took place in three steps. In the first step, two researchers (F. Diotallevi, O. Simonetti) independently selected the articles based on the title. Any disagreement was solved by consulting a senior investigator (A. Offidani). The second step consisted of evaluating the abstracts. At least three members of the research team (G. Rizzetto, G. Radi, E. Martina) independently assessed each abstract. The research team resolved all discrepancies through consensus. Linguistic revision was performed by E. Martina.

### 2.4. Document Results

All sources with similar data/level of evidence were analyzed, collected, and grouped. The main text was structured into subsections. New evidence-based points were summarized and major points for future research and practice were defined.

## 3. Results

Two macro topics emerged from the literature review: “pathophysiology of pediatric psoriasis” and “the use of biologic drugs in pediatric psoriasis”.

### 3.1. Pathophysiology of Pediatric Psoriasis

The pathogenesis of pediatric psoriasis involves a complex interaction between the immune system and the two main factors that influence the onset and exacerbation of the disease: environmental factors and genetic predisposition. Approximately one-third of patients with psoriasis have a first-degree relative with the condition, emphasizing the influence of genetics [13,14].

Environmental factors are involved in the expression of the disease. The most frequent of these include: high stress when reaching the age to start school, detachment from parents, mild mechanical trauma (Koebner phenomenon), drugs and infections [15]. Regarding systemic drugs as trigger factors (lithium, beta blockers, antimalarial agents, etc.) or HIV infection, they are less frequently implicated in children than in adults [5]. Regarding infections, however, group A streptococcus (GAS) throat infection is the most frequent trigger for pediatric psoriasis, being mainly responsible for guttate psoriasis, which can then lead to chronic plaque psoriasis [5].

Pediatric psoriasis patients exhibited a distinct difference in the expression of interleukin (IL)-17 and IL-22 compared to that of healthy pediatric controls and adult psoriasis patients [16]. Indeed, in adults, inflammation is driven predominantly by type 17 T helper (Th17)/IL-23 pathway activation with a positive inflammatory feedback loop due to the persistent release of proinflammatory cytokines, particularly Tumor Necrosis Factor (TNF)-α, IL-23, and IL-17A. In children, however, it is reported in the literature that the lesional tissue of patients with pediatric psoriasis is associated with higher levels of TNF-α, IL-22-producing T cells and relatively fewer IL-17-producing T cells compared with adult psoriasis [16,17,18]. In addition, Zhang et al. reported that increased Th17 and regulatory T cells in the peripheral circulation correlated positively with disease severity [17]. Therefore, distinct immunophenotypic findings in pediatric psoriasis, the differences between pediatric and adult psoriasis, and between individual patients, may be useful in determining targeted and more personalized therapies for pediatric psoriasis. However, previous studies on this topic were small pilot studies with small sample sizes [16,17,18]. Thus, larger studies need to be conducted to investigate these specific immunophenotypic targets for pediatric psoriasis.

### 3.2. Biologic Treatments of Pediatric Psoriasis

Biologic drugs have revolutionized the treatment of moderate to severe plaque psoriasis due to their high efficacy and relative side effects.

Several biologic agents have recently been approved for use in pediatric psoriasis. Compared with other systemic therapies, which are commonly used off-label in children, biologics are very convenient because they have better dosing protocols and require less laboratory testing. In addition, because they are targeted drugs, the risk of systemic toxicity may be lower [19]. There are currently five biologic drugs approved for the treatment of pediatric psoriasis (adalimumab, etanercept, ustekinumab, secukinumab, ixekizumab) and six drugs currently undergoing phase 3 studies (brodalumab, guselkumab, risankizumab, tildrakizumab, certolizumab pegol, deucravacitinib) [20]. Biologic drugs approved for use in pediatric psoriasis are shown in Table 1.

#### 3.2.1. Anti TNF-α Drugs

TNF-α inhibitors have been used to treat rheumatoid arthritis, TNF 1-associated fever, juvenile idiopathic arthritis, ulcerative colitis, and Crohn’s disease in children for more than a decade [21]. Data on TNF-α blockers suggest that they are safe and effective in managing pediatric psoriasis. There are four drugs currently used for adult psoriasis (adalimumab, certolizumab pegol, etanercept and infliximab) and only two (adalimumab and etanercept) are approved for use in children.

A phase 3 clinical trial is currently underway to evaluate the efficacy and safety of certolizumab pegol in the treatment of moderate to severe chronic plaque psoriasis in study participants aged 6 to 11 and 12 to 17 years [22], but no data are available yet.

Adalimumab

Adalimumab is a fully human anti TNF-α monoclonal antibody. In Europe, adalimumab has been approved for the treatment of psoriasis in children and adolescents (aged ≥ 4 years) who have had an insufficient reaction to topical therapy and phototherapy. While adalimumab is FDA-approved for children with Crohn’s disease (6 years or older) and Juvenile Idiopathic Arthritis (JIA) (2 years or older), it is not approved for pediatric psoriasis indication in the USA [23,24]. Adalimumab showed good safety and efficacy in pediatric psoriasis. A phase 3 study compared adalimumab and methotrexate in 114 pediatric patients aged 4–17 years with severe plaque psoriasis. At week 16, 58% of subjects receiving adalimumab 0.8 mg/kg-full dose achieved PASI (Psoriasis Area Severity Index) 75 versus 32% of those taking oral methotrexate (MTX) 0.1–0.4 mg/kg/week [23,25]. In a retrospective, multicenter study over a 52-week period, efficacy and the response in biologic-naïve versus non-naïve patients were observed. At week 16, 29.6% of patients achieved a 90% PASI score reduction, while 55.5% of patients achieved a 75% PASI score reduction, and effectiveness was sustained through week 24. Moreover, the PASI response rates did not differ between biologic-naïve and non-naïve patients [26].

Infection was one of the common side effects, but there was no significant difference among the groups [23,26].

Etanercept

Etanercept is a recombinant protein that blocks TNF- α from binding to its receptor [27]. It is the TNF-α inhibitor most extensively studied in children and in the United States. It is indicated for moderate to severe polyarticular juvenile rheumatoid arthritis and pediatric plaque psoriasis in patients aged 4 years and older [28,29]. In the European Union, in children, etanercept is indicated to treat juvenile idiopathic arthritis, polyarthritis (rheumatoid-factor-positive or -negative), extended oligoarthritis and chronic severe pediatric plaque psoriasis in children over the age of six [30].

In 2008, 211 patients with moderate to severe plaque psoriasis aged 4 to 17 years were treated for 12 weeks with etanercept at a dosage of 0.8 mg/kg or placebo, followed by an open-label duration of 24 weeks and a second randomization to 36 weeks to examine the effects of treatment discontinuation in a randomized, double-blind, phase 3 clinical trial [31]. PASI 75 and PAI 90 were found in 57% and 27% of children, respectively, at the end of the first 12-week period, compared with 11% and 7% in the placebo group. This benefit appears to be maintained in the majority of patients after 96 weeks [32]. Etanercept is associated with fewer AEs than methotrexate or acitretin [33]. In a study of 181 patients who received long-term etanercept, 145 (80.1%) patients experienced adverse events, such as upper respiratory tract infections (24.9%), nasopharyngitis (17.1%), streptococcal pharyngitis (12.7%), headache (11.6%), and sinusitis (10.5%). During the follow-up period, there were no cases of opportunistic infections or malignancy reported in children or adolescents [32,34].

Infliximab

Infliximab is a chimeric monoclonal antibody that binds to human TNF-α and prevents it from binding to its receptor in both soluble and membrane-bound forms [35]. Many countries have approved it for use in adults with psoriasis and other diseases [35]. It has been approved by the FDA for use in children aged 6 years and older who have Crohn’s disease or ulcerative colitis, and several cases of reduced psoriasis have been reported after a few weeks of therapy [36,37,38]. However, it is not approved for the treatment of pediatric psoriasis in any country [39].

#### 3.2.2. Anti-IL-12/23 and Anti-IL-23

Another class of drugs widely used for the treatment of plaque psoriasis is those that target interleukins 12 and 23, cytokines that are at the top of the inflammatory cascade that causes psoriasis. There is currently only one drug approved for pediatric use, which targets both IL-12 and IL-23 (ustekinumab). Three other drugs, guselkumab, risankizumab and tildrakizumab, are currently in phase 3 trials for pediatric use [20].

Ustekinumab

Ustekinumab is a fully human monoclonal antibody targeting the IL-12 and IL-23 axis and is efficient for psoriasis and psoriatic arthritis. Specifically, it binds to the p-40 subunit of both IL-12 and IL-23 so that they subsequently cannot bind to their receptors [40].

Ustekinumab is FDA- and EMA-approved for treatment of moderate to severe psoriasis in children aged 6 years and older [41,42]. Dosing for ustekinumab is weight-based. Children <60 kg should receive 0.75 mg/kg, children between 60 kg and 100 kg should receive 45 mg, and children >100 kg should receive 90 mg. Loading doses of ustekinumab are administered at weeks 0 and 4, and then maintenance dosing is administered every 12 weeks.

In the phase 3 CADMUS Jr study, patients with moderate to severe plaque psoriasis, aged 6 years to less than 12 years, received ustekinumab. After 12 weeks, 77% of patients achieved a Physician’s Global Assessment (PGA) 0/1 response. In addition, 84% of patients achieved a PASI 75 response and 64% achieved a PASI 90 response [41]. The results of this study overlapped with those of a previous phase 3 study (CADMUS) that established the efficacy and safety of ustekinumab in patients over 12 years of age [42].

The CADMUS and CADMUS Jr clinical trials showed that ustekinumab was well-tolerated in children with plaque psoriasis. The most common AEs were nasopharyngitis, upper respiratory tract infection and pharyngitis. No cases of opportunistic infections or malignancy were reported [41,42].

Guselkumab

Guselkumab is a fully human IgG1λ monoclonal antibody that selectively targets the unique p19 subunit of human IL-23 without binding IL-12. It is approved for the treatment of plaque psoriasis in adults by both the FDA and EMA [43,44,45,46,47]. In adult patients with plaque psoriasis, guselkumab demonstrated superior clinical responses and was better tolerated compared with adalimumab and ustekinumab [43,46]. In addition, two other clinical trials demonstrated the non-inferiority of guselkumab to anti-IL-17 secukinumab and ixekizumab in achieving skin clearance in adult patients with plaque psoriasis [48,49]. The safety profile of guselkumab was similar to that of ustekinumab, having a slightly higher incidence of adverse events, with infection being reported most often [46].

Presently, no study on the efficacy and safety of the IL-23 antibody for pediatric patients has been published. A study is now being conducted (NCT03451851) to evaluate the efficacy and safety of guselkumab in relation to etanercept and placebo in pediatric patients with chronic plaque psoriasis [50].

Risankizumab

Risankizumab is an IgG1k monoclonal antibody that, like guselkumab, binds the p-19 subunit of IL-23 and is the latest anti-IL-23 drug approved for the treatment of moderate to severe plaque psoriasis in adults. The latest head-to-head phase 3 trial and a recent meta-analysis provided further evidence of the superior efficacy of risankizumab compared to adalimumab for treatment of moderate to severe psoriasis [51,52]. A study of subcutaneous risankizumab injection in relation to ustekinumab for pediatric participants with moderate to severe psoriasis is underway (NCT04435600) [53].

Tildrakizumab

Tildrakizumab is an IgG1k monoclonal antibody that, like the other two anti-IL-23, binds the p19 subunit of the IL-23. Among the IL-23 p19 inhibitors and also in comparison to IL-17 inhibitors, tildrakizumab seems to have slightly lower efficacy in terms of PASI scores for the treatment of plaque psoriasis in adult patients [52]. Pooled data from RCTs revealed that patients who would achieve a PASI > 90 response at week 28 could be identified as early as week 4 and they maintained their week 28 PASI improvement at week 52 [52,54]. A study of subcutaneous tildrakizumab injection for pediatric participants with moderate to severe psoriasis is underway (NCT03997786).

#### 3.2.3. Anti-IL-17

IL-17A is considered, along with IL-23, the most important target of plaque psoriasis therapy and has been recognized as the critical effector cytokine in psoriasis. Presently, the three approved IL-17 pathway inhibitors are secukinumab and ixekizumab, both IL-17A inhibitors, and brodalumab, which targets the IL-17-receptor A (IL-17RA). Several RCTs, sustained by real-life data, proved the efficacy and safety of these treatments, showing higher rates of complete or almost complete clearance compared to TNF-alfa inhibitors and ustekinumab [48,49]. Secukinumab and ixekizumab were approved by the EMA and FDA as new treatment options for moderate to severe psoriasis in patients aged 6–17 years [55,56], while brodalumab is under clinical study [57].

Secukinumab

Secukinumab is a fully human IgG1κ monoclonal anti-IL-17A antibody. It is currently approved by the FDA and EMA for the treatment of pediatric plaque psoriasis in children up to 6 years old. The phase 3 study that enabled the drug’s approval showed that both doses of secukinumab, a low dose and a high dose, had superior efficacy compared with placebo in PASI 90 response (72.5% and 67.5% vs. 2.4%), PASI 75 response (80.0% and 77.5% vs. 14.6%) and IGA at week 12.

The response trend was maintained until week 52 (PASI 75/90/100: low-dose secukinumab, 87.5%/75.0%/40.0% and high-dose secukinumab, 87.5%/80.0%/47.5% vs. etanercept, 68.3%/51.2%/22.0% and IGA 0 or 1: low-dose secukinumab, 72.5% and high-dose secukinumab, 75.0% vs. etanercept, 56.1%). Safety profiles were similar to those in adult studies with secukinumab: the most common (incidence ≥ 10%) AEs reported up to week 52 were nasopharyngitis (23.5% with any low dose of secukinumab and 26.0% with any high dose of secukinumab vs. 26.8% with etanercept) and headache (10.2% and 10.0% vs. 9.8%) [56].

Ixekizumab

Ixekizumab is a humanized IgG4 monoclonal antibody that selectively binds to IL-17A, approved by the FDA and EMA for the treatment of adults and children six years and older with moderate to severe plaque psoriasis. The double-blind, randomized, placebo-controlled phase 3 IXORA-PEDS study addressed the high-affinity monoclonal antibody to IL-17A ixekizumab in pediatric psoriasis [55]. Ixekizumab was superior to placebo in PASI 75 (89% vs. 25%) at week 12. Ixekizumab also showed superior results in PASI 75 and PGA at week 4, improved quality of life, and complete skin clearance. The safety profile of ixekizumab was generally consistent with that of adults with moderate to severe plaque psoriasis [55]. In addition, a new citrate-free formulation of ixekizumab has recently been made available, including the same active ingredient as the original formulation. This formulation has significantly reduced injection site pain felt by some people immediately after injection, as demonstrated by an 86% reduction in visual analog scale (VAS) pain compared with the original formulation [58].

Brodalumab

Brodalumab, a recombinant fully human monoclonal IgG2 antibody with high affinity to human interleukin (IL)-17RA, is approved for the treatment of moderate to severe plaque psoriasis in adults. The EMBRACE study, a phase 3 trial that aims to investigate the efficacy and safety compared with placebo, and the safety compared with ustekinumab of brodalumab in adolescents with moderate to severe plaque psoriasis, is currently underway (NCT04305327). The study will also evaluate whether brodalumab affects the development of vaccination-induced immune responses [57].

Phase 3 studies performed in adults have fully demonstrated the efficacy and safety of the drug in the treatment of plaque psoriasis [59]. Consistently, >80% of patients achieved PASI-75, and efficacy was maintained for >2 years. The high affinity of brodalumab to human IL-17RA blocks the biological activities of the pro-inflammatory cytokines IL-17A, IL-17C, IL-17E, IL-17F, and IL17A/F heterodimer, resulting in inhibition of the inflammation and clinical symptoms associated with psoriasis. This mechanism of blocking multiple IL-17 family cytokines differs from that of other available biologics which selectively target some parts of the Th-17 axis and may account for the effectiveness of brodalumab in patients poorly responsive to other biologics, a feature which has also been shown where subgroup analysis has been undertaken in clinical trials. The drug is well-tolerated during the normal 12-week induction phase and with prolonged treatment (52 to 120 weeks) [10].

## 4. Discussion

Pediatric psoriasis is a common disorder that often begins in childhood. In a questionnaire survey of 5600 individuals with psoriasis (median age 44 years, range 1 to 92 years), approximately one-third reported onset of disease during the first two decades of life and 10% reported onset prior to the age of 10 years [4]. The major clinical forms of psoriasis are similar in children and adults and chronic plaque psoriasis is the most common form of psoriasis in children [6,60,61].

In addition, children with psoriasis are more likely to present with psoriasis involving the face, scalp, or intertriginous skin than adults, and it is known that involvement of these areas affects the patient’s DLQI and requires urgent treatment [6].

Pediatric psoriasis is associated with an increased risk of various comorbidities, including obesity and other cardiovascular risk factors, arthritis, and psychiatric disorders [8].

Obesity and metabolic syndrome appear to be the most common comorbidities of pediatric psoriasis. Population-based studies have found increased rates of hyperlipidemia and lipid peroxidation, hypertension and diabetes in psoriatic children [62,63,64,65].

However, although psoriasis is independently associated with an increased risk of metabolic syndrome, this risk appears to be largely driven by obesity [65].

All of this makes adequate treatment of pediatric psoriasis critical. Currently, biological therapy appears to be the only systemic therapy, besides phototherapy, approved for pediatric use. Other treatments, such as methotrexate and cyclosporine, although they have been used in children for years, in fact have an off-label indication and are not without side effects [20].

The literature and clinical practice tend to increase the use of biologics, with 25% of children with psoriasis treated with these drugs [66]. In determining which patients will benefit from biologic therapy, as in adults, children and adolescents with BSA greater than 10%, DLQI > 10, failed standard therapies, or other contraindications should be considered [67]. Some studies suggest using systemic therapies earlier in mild psoriasis and in patients with lower DLQI scores. Allowing patients to switch to systemic treatment may improve quality of life and reduce the progression of the disease and its comorbidities [33].

Moreover, biologic drugs, compared with other systemic treatments such as methotrexate and cyclosporine, require less rigid treatment schedules, have fewer adverse effects, less need for monitoring, and greater efficacy [33]. However, the long-term safety profile of these drugs is still being defined [67]. In addition, vaccinations are another special consideration for patients starting biologic therapy. Live vaccines are absolute contraindications to biologic therapy [33,68,69,70,71].

Currently, there are several biologic drugs approved for use in pediatric psoriasis. In addition to anti-TNF-α, of which etanercept certainly appears to be the most studied with regard to safety in the pediatric population, drugs targeting IL-17 (secukinumab and ixekizumab) are also approved to date, and other drugs such as brodalumab, which targets the IL-17 receptor, and anti-IL-23 drugs are under investigation. The efficacy of these drugs appears to be non-inferior to anti-TNF alpha, and, as far as currently approved drugs are concerned, the safety profile is similar [20]. Current available guidelines do not indicate which biologic should be used as the first-line treatment. In our opinion, given the extensive literature demonstrating its high safety profile, etanercept might be indicated as first-line treatment [72]. Further studies, including comparative studies with other biologics, may better indicate the treatment course.

## 5. Conclusions

Psoriasis in children poses a number of challenges. It has age-specific clinical features, the appearance of which can change over time. Most treatment modalities for adult psoriasis have been used off-label for pediatric psoriasis. Although studies have been limited, recent research has proposed a difference in TNF-α, IL-17, IL-22 expressions in pediatric psoriasis patients compared with those in adult psoriasis patients. Biologics, as targeted therapy, are also gradually being used for pediatric psoriasis with great efficacy and a similar safety profile as adults. It is reasonable to assume that a more targeted therapeutic approach for pediatric psoriasis will be guided by the unique immunophenotypic characteristics of psoriasis in children, characteristics that are not fully known and need to be better investigated. Thus, we believe that further studies are needed to identify the different pathogenetic processes in pediatric psoriasis.

## Figures and Tables

**Table 1 ijms-23-11128-t001:** Biologics approved for pediatric psoriasis.

Biologics	Mechanism	FDA/EMA Approval	Dosing	Safety
Adalimumab	Fully human anti TNF-α monoclonal antibody	Approved by EMA for children ≥ 4 years of age	Patient weight ≥ 15 kg and < 30 kg: initial dose of 20 mg SC, followed by 20 mg SC every other week	Upper respiratory infections, uncomplicated infections, injection site reactions
			Patient weight ≥ 30 kg: initial dose of 40 mg SC, followed by 40 mg SC every other week	
Etanercept	A fusion protein blocking TNF-α from binding to its receptor	Approved by FDA for treatment of psoriasis in patients ≥ 4 years of age	25 mg twice weekly or 50 mg SC once weekly	Upper respiratory tract infections, nasopharyngitis, streptococcal pharyngitis, sinusitis, headache, injection site reactions
		Approved by EMA for treatment of psoriasis in patients ≥ 6 years of age	Alternatively: 50 mg SC twice weekly for up to 12 weeks, followed by 25 mg SC twice weekly, or 50 mg SC once weekly, if needed	
Ustekinumab	A monoclonal antibody that targets the p40 subunit of IL-12 and IL-23	Approved by FDA/EMA for treatment of psoriasis inpatients ≥ 6 years of age	Administer at week 0 and 4, and then every 12 weeks	Upper respiratory tract infection, headache, injection site reaction
			Body weight < 60 kg: 0.75 mg/kg SC (table with injection volumes available in EMA label)	
			Body weight ≥ 60 kg SC and ≤ 100 kg: 45 mg SC	
			Body weight > 100 kg: 90 mg SC	
Secukinumab	A fully human, monoclonal anti-IL-17A antibody	Approved by FDA/EMA for treatment of psoriasis in patients ≥ 6 years of age	Weekly dosing for 5 weeks, followed by monthly dosing for maintenance	Upper respiratory tract infection, headache
			Body weight < 25 kg: 75 mg SC	
			Body weight ≥ 25 kg and < 50 kg: 75 mg SC	
			Body weight ≥ 50 kg: 150 mg SC (may be increased to 300 mg SC)	
Ixekizumab	A humanized, monoclonal anti-IL-17A antibody	Approved by FDA/EMA for treatment of psoriasis in patients ≥ 6 years of age (with body weight ≥ 25 kg)	Patient weight 25–50 kg: start with 80 mg SC, followed by 40 mg SC every 4 weeks	Upper respiratory tract infection, injection site reaction, bronchitis and sinusitis
			Patient weight > 50 kg: start with 160 mg SC (two 80 mg-injections), followed by 80 mg SC every 4 weeks	

## Data Availability

Not applicable.

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
