# Peer review of "Biological Treatments for Pediatric Psoriasis: State of the Art and Future Perspectives"

_ijms, 2022, doi:10.3390/ijms231911128_

Round 1

Reviewer 1 Report

I was pleased to read this interesting  review on biological treatments for pediatric psoriasis. The manuscript is clearly structured and cites the most recent literature relevant to the topic.

However, to further improve their manuscript, I suggest the authors to consider my following comments:

Ø  wich molecule should be considered a first line treatment, if there is one.

Ø  the language is of good quality but there are several spelling mistakes that should be carefully checked by the authors.

Ø  are there biologic options for the other forms of psoriasis in children?

Ø  are there any differences in the expressions of TNF-α-IL-17-IL-22 in pediatric psoriasis patients compared to those of adult psoriasis patients?

Ø   almost all references are not writen correctly (1,2 ??? ).

Ø  I also believe that a phrase should be added in which the authors compare all the therapeutic options in a more comparative way.

Despite the title suggests that the article sought to investigate “future perspective” , the authors did not discuss this issue more clearly 

Author Response

Dear Editor and reviewers,

first of all, I would like to thank you for the review and for the valuable comments that allowed us to improve the manuscript. Here is the revision point by point. All changes made have been highlighted in yellow in the text. For convenience, reference numbers, inserted in the text, are highlighted in red.

Reviewer 1

I was pleased to read this interesting review on biological treatments for pediatric psoriasis. The manuscript is clearly structured and cites the most recent literature relevant to the topic.

However, to further improve their manuscript, I suggest the authors to consider my following comments:

Q: which molecule should be considered a first line treatment, if there is one.

A: Thank you for your comment. That part has been added to the discussion.

Q:  the language is of good quality but there are several spelling mistakes that should be carefully checked by the authors.

A: A correction of spelling errors was made.

Q:  are there biologic options for the other forms of psoriasis in children?

A: Currently there are no approved treatments for other forms of psoriasis, although there are case reports of efficacy in the literature. However, to avoid confusion, this review focused only on the treatment of plaque psoriasis.

Q:  Are there any differences in the expressions of TNF-α-IL-17-IL-22 in pediatric psoriasis patients compared to those of adult psoriasis patients?

A: Thank you for your comment. The differences in cytokine expression in pediatric and adult psoriasis are explained in the section "pathophysiology of pediatric psoriasis." These differences may play a role in the future in the development and prediction of the efficacy of targeted biologic therapies for pediatric psoriasis but as yet there are few studies. All of this is better explained in the section.

Q:   almost all references are not written correctly (1,2 ??? ).

A: References have been corrected.

Q:  I also believe that a phrase should be added in which the authors compare all the therapeutic options in a more comparative way.

A: This sentence has been added to the discussion.

Q: Despite the title suggests that the article sought to investigate “future perspective” , the authors did not discuss this issue more clearly.

A: Future perspectives regarding the selection of targeted treatments based on disease characteristics are discussed in the discussion and conclusions.

Reviewer 2

Q: This is a good and important review

A: Thank you.

Reviewer 3

Thanks for submitting your review on pediatric psoriasis.

Minor queries for p. 2:

Q: In the first step, three researchers (F.D., O.S.) were involved." Who was the the third?

A: The error has been corrected as two researchers were involved.

Q: Please use either abbreviations (.S.) or full names (G. Rizzetto etc.).

A: Full surnames have been given to avoid misunderstandings (example, E. Molinelli and E. Martina).

Q: p 6. Table, Izekizumab: "Upper respiratory tract infection".

A: Done.

Q: p. 14: Izekizumab is doubled.

A: Done.

Reviewer 2 Report

This is a good and important review

Author Response

Dear Editor and reviewers,

first of all, I would like to thank you for the review and for the valuable comments that allowed us to improve the manuscript. Here is the revision point by point. All changes made have been highlighted in yellow in the text. For convenience, reference numbers, inserted in the text, are highlighted in red.

Reviewer 1

I was pleased to read this interesting review on biological treatments for pediatric psoriasis. The manuscript is clearly structured and cites the most recent literature relevant to the topic.

However, to further improve their manuscript, I suggest the authors to consider my following comments:

Q: which molecule should be considered a first line treatment, if there is one.

A: Thank you for your comment. That part has been added to the discussion.

Q:  the language is of good quality but there are several spelling mistakes that should be carefully checked by the authors.

A: A correction of spelling errors was made.

Q:  are there biologic options for the other forms of psoriasis in children?

A: Currently there are no approved treatments for other forms of psoriasis, although there are case reports of efficacy in the literature. However, to avoid confusion, this review focused only on the treatment of plaque psoriasis.

Q:  are there any differences in the expressions of TNF-α-IL-17-IL-22 in pediatric psoriasis patients compared to those of adult psoriasis patients?

A: Thank you for your comment. The differences in cytokine expression in pediatric and adult psoriasis are explained in the section "pathophysiology of pediatric psoriasis." These differences may play a role in the future in the development and prediction of the efficacy of targeted biologic therapies for pediatric psoriasis but as yet there are few studies. All of this is better explained in the section.

Q:   almost all references are not written correctly (1,2 ??? ).

A: References have been corrected.

Q:  I also believe that a phrase should be added in which the authors compare all the therapeutic options in a more comparative way.

A: This sentence has been added to the discussion.

Q: Despite the title suggests that the article sought to investigate “future perspective” , the authors did not discuss this issue more clearly.

A: Future perspectives regarding the selection of targeted treatments based on disease characteristics are discussed in the discussion and conclusions.

Reviewer 2

Q: This is a good and important review

A: Thank you.

Reviewer 3

Thanks for submitting your review on pediatric psoriasis.

Minor queries for p. 2:

Q: In the first step, three researchers (F.D., O.S.) were involved." Who was the the third?

A: The error has been corrected as two researchers were involved.

Q: Please use either abbreviations (.S.) or full names (G. Rizzetto etc.).

A: Full surnames have been given to avoid misunderstandings (example, E. Molinelli and E. Martina).

Q: p 6. Table, Izekizumab: "Upper respiratory tract infection".

A: Done.

Q: p. 14: Izekizumab is doubled.

A: Done.

Reviewer 3 Report

Thanks for submitting your review on pediatric psoriasis. 

Minor queries for p. 2::

1. "In the first step, three researchers (F.D., O.S.) were involved." Who was the the third?

2. Please use either abbreviations (.S.) or full names (G. Rizzetto etc.).

p. 6. Table, Izekizumab: "Upper respiratory tract infection".

p. 14: Izekizumab is doubled.

Author Response

(The authors gave the same response as above.)
